# Spatial patterns and predictors of antenatal care interruption in war-torn Tigray, northern Ethiopia: Spatial modelling approach

Assefa Ayalew Gebrselassie[1]*, Mussie Alemayehu[1,2], Haftu Gebrehiwot[2], Brhane Ayele[2], Hailay Gebretnsae[2], Ferehiwot Hailemariam[1], Tsegay Wellay[1], Adhena Ayalew[1], Araya Abrha Medhanyie[1,3], Znabu Hadush Kahsay[1], Liya Mamo[1], Mebrahtu Kalayu Chekole[1], Reda Shamie[1], Mohammedtahir Yahya[4,5], Melaku Abraha[5], Hayelom Kahsay[2], Tsegay Hadgu[2], Fana Gebreslassie[2], Asfawosen Aregay[2], Kiros Demoz[2], Mulugeta Tilahun[2], Mulugeta Woldu[2], Ataklti Gebrtsadik[2], Abraham Aregay Desta[2], Gebrehaweria Gebrekuristos[6], Amanuel Haile[6], Rieye Esayas[6], Tsegay Berihu[6], Abrham Gebrelibanos[3], Tadele Tesfean[3], Ashenafi Asmelash[7], Afework Mulugeta Bezabh[1]

1 School of Public Health, Mekelle University, College of Health Sciences, Mekelle, Ethiopia, 2 Tigray Health Research Institute, Mekelle, Tigray, Ethiopia, 3 MARCH research center, College of Health Sciences, Mekelle, Tigray, Ethiopia, 4 Department of Obstetrics and Gynecology, School of Medicine, Mekelle University, College of Health Sciences, Mekelle, Ethiopia, 5 Mekelle Hamlin Fistula Center, Mekelle, Tigray, Ethiopia, 6 Tigray Health Bureau, Mekelle, Tigray, Ethiopia, 7 Mums for Mums, Mekelle, Tigray, Ethiopia

* aayalew120@gmail.com

## Abstract

### Introduction

Antenatal care (ANC) is vital for maternal health, yet its provision has been severely disrupted in conflict-affected Tigray, Ethiopia. This study employs a community-based approach to analyze the spatial patterns and predictors of ANC interruption in conflict-affected, Tigray region, northern Ethiopia.

### Methods and materials

A cross-sectional design was used, to select 2,444 eligible women from 13,915 women of reproductive age across from six zones and 57 clusters. Multistage cluster sampling was employed to approach the study participants. Data were collected using structured questionnaires, and geographic coordinates were obtained for spatial geo-points. A spatial logistic regression model was used to assess the effects of geographic and socio-demographic factors on ANC interruptions, accounting for spatial dependencies.

**Data availability statement:** Data is available within the manuscript.

**Funding:** UNICEF, WHO, Amref Health Africa and UNFPA and Tigray regional heal bureau and Mums for Mums (MfM) for all rounded support in ensuring a successful and productive integrated research data collection process we assure there is no any competing programming and any other support that influences the research output. The funders had no role in study design, data collection and analysis, decision to publish, or preparation of the manuscript.

**Competing interests:** The authors have declared that no competing interests exist.

## Results

The prevalence of ANC interruption was 24.8% (n = 605, 95% CI: 23.1%, 26.5%). Each additional minute of travel increased the log-odds of interruption by 0.000353 (SE = 0.0001, p < 0.001. Geographic zone significantly influenced ANC interruptions, with women in the southeast and eastern zones experiencing lower log-odds compared to the northwest zone. Moreover, having more under- five children raised the log-odds of ANC interruption by 0.083 (SE = 0.01, p < 0.001). Households without a radio had a 0.089 lower log-odds of ANC interruption (SE = 0.033425, p = 0.007), highlighting the importance of access to information. The war's impact was significant, with 96.5% (2,358, 95% CI: [95.4%, 97.6%]) reporting a family member was disabled; of these, 72.7% (1,778, 95% CI: [71.0%, 74.4%]) did not postpone their ANC visit, while 23.7% (580, 95% CI: [22.2%, 25.2%]) did.

## Conclusions

The findings revealed a significant ANC interruption in Tigray, influenced by geographic disparities, household dynamics, and information access. Improving infrastructure and security, implementing targeted interventions, enhancing mass media health information dissemination, and further research on geographic disparities are essential.

## Introduction

Antenatal care (ANC) is essential for the health of pregnant women and their fetuses, aiding in labor preparation and the recognition of pregnancy-related danger signs [1,2]. ANC interruption is defined as not receiving ANC care as per the new World Health Organization (WHO) recommendations, which states that pregnant mothers should attend at least eight antenatal care (ANC) visits throughout their pregnancy [2]. In conflict-affected populations, women and children face disproportionate morbidity [3].Evidence from Syria indicates that the prenatal care coverage, which includes one visit with a trained attendant, has decreased [4]. Women in conflict-affected populations suffer differently of difficulties in obtaining reproductive healthcare, higher rates of gender-based violence, and unfavorable pregnancy outcomes, which are made worse by stress and a lack of medical resources [5,6].

A survey conducted in Eastern Burma found that 88% of women reported giving birth at home during their most recent pregnancy, that ANC visits were 16.7% and skilled birth was just 5.1%. The survey also revealed that women were often exposed to human rights violations [6]. From 2000 to 2016 as of the EDHS, the proportion of Ethiopian women receiving prenatal care (ANC) from a competent provider increased from 27% to 34% and 62%, respectively with the Tigray region having the highest rate at 90% [7]. Another study conducted in Tigray showed that women whose husbands arranged transportation for ANC had over twice the odds of being alive compared to those whose husbands did not [8]. According to the study's analysis of

the health system of Tigray during the Tigray war showed that, the devastation of Tigray's health system was deliberate, systematic, and pervasive, impacting all tiers of the health system zone [9]. According to a recent community-based study, maternal mortality increased from 186 in 2019 per 100,000 live births to 840 per 100,000 live births in 2023 after the Tigray War [10].

Studies indicate that armed conflict has a significant negative impact on maternal healthcare consumption. For instance, research conducted in Uganda, a region severely affected by conflict, shows that access to maternal healthcare diminishes [11]. A recent study conducted in Ethiopia's Tigray region, the health system trends are in line with war events. The analysis of the region's zones revealed that the Mekelle and North Western zones were the most stable, while the remaining zones were the least stable, with many missing data points pointing to instability [12]. Similarly, a prospective study during the war in Lebanon revealed that women were neglecting their usual prenatal care, which resulted in an increase in health problems during that period [13]. Additionally, a study focusing on Northern Uganda and Burundi found that armed conflict adversely affected the health system, leading to reduced access to and quality of maternal and reproductive health services [14].

The 2019 Ethiopian Demographic Health Survey indicated that 74% of pregnant women nationally received antenatal care (ANC) from skilled providers. In the Tigray region, this percentage was significantly higher at 94% [15]. However, a study conducted during the Tigray conflict found that in the first 90 days of the war, there were no ANC services available. Additionally, only 27.5% of hospitals, 17.5% of health centers, and 11% of ambulances were operational, while none of the 712 health posts were functional [16].

Understanding the spatial patterns of ANC interruption—identifying specific areas within Tigray with varying interruption rates—and the predictors of these interruptions, such as conflict impact, and socioeconomic factors remain limited. Therefore, assessing the percentage of women with ANC interruptions and their peculiar factors is crucial for developing effective maternal health interventions in Tigray post-conflict, northern Ethiopia.

## Methods and materials

### Study setting, design, data source

This cross-sectional study is one of the integrated surveys aimed at evaluating cost-effectiveness in post-conflict settings on identifying the behavioral and social drivers of COVID-19 vaccination, maternal and child health services in Tigray, Ethiopia. The study involved 13,915 female participants, of whom 10,654 were of reproductive age (15–49 years), and 3,933 had been pregnant since the onset of the conflict. The analysis focused on 2,444 women who were pregnant women during the conflict had received antenatal care (ANC). Data collection occurred over a month, from August 1–30, 2023 [17]. This systematic approach allowed for a comprehensive examination of factors affecting interruption in ANC within this vulnerable population.

**Study design.** Community based cross sectional study

### Measurements and variables

**Dependent variable: Antenatal care interruption.** Measured as a binary variable indicating whether ANC services were interrupted (1) or not interrupted (0) during the conflict period. This was assessed through interviews with the participants in selected households of mothers who were following ANC visit during the conflict and interrupted their ANC visits for different reasons.

**Independent variables.** Independent variables include a range of socio-economic, geographical, and conflict-related factors:

**Socio-economic factors:** Age of mother, education level, occupation, marital status, number of children, media access, ownership of radio, mobile and television.

**Geographical factors:** Zone of residence, distance to health facility: Distance to nearest health facility was measured in and it was the time taken to double trip from the home to the nearest health facility providing ANC services. Accessibility indexes a composite measure reflecting road conditions, transportation availability, and geographic barriers, security. Conflict-related factors; displacement status whether the mother or more households have been displaced due to conflict, and it was labeled as (yes/no).Perceived security was measured a qualitativelybased on feelings of safety categorized as Yes or No when accessing health services. Loss to property and close family member who care for them were measured as Yes or No.

**Data collection methods:** Structured questionnaires interviewer-administered were used to collect data from expectant mothers to gather data on ANC utilization, socio-economic status, and perceived barriers. Geographic positioning Systems (GPS) data were collected through the ODK tool using mobile tablets and the X-Y coordinates of the household were recorded.

## Statistical method and analysis

Univariate descriptive, bivariate, chi-square and multivariate spatial logistic regression approaches were employed to analyze the ANC data from targeted zones of the Tigray Region. The univariate analyses were used to summarize and describe the demographic variables (e.g., age, gender, etc.), using frequencies, percentages, means, and standard deviations provided a detailed overview. Associations between categorical variables (e.g., gender, zone, etc.) and ANC interruptions were assessed using Chi-square or Fisher's exact tests. We used ANC interruption to model the relationship between risk factors and occurrence across zones. Spatial logistic regression is an advanced modelling approach that incorporates zones, representing administrative or geographic divisions, to analyze binary outcomes while accounting for spatial dependencies. The model extends ordinary logistic regression by including spatial random effects ($\varnothing_z$) to capture unobserved spatial variation between zones, improving the accuracy of estimates and inference. The model is expressed as:

$$\text{logit}\,(P_i|)) = \ln\left\{\frac{P_i}{1-P_i)}\right\} = \beta_o + \sum_{k=1}^{p} \beta_k X_{ik} + \varnothing_z, \quad z = S, SE, M, E, C, NW$$

where $P_i$ represents the probability of the outcome for observation $i$, $\beta_o$ is the intercept, $X_{ik}$ are predictors, $\beta_k$ are their coefficients, and $\varnothing_z$ captures the spatial effect for zone $z$ This approach allows the model to account for spatial clustering and patterns, addressing biases and inaccuracies that arise from spatial autocorrelation in ordinary logistic regression. Spatial autocorrelation in the dependent variables was assessed using Moran's I test. The spatial logistic regression model, which incorporates structured spatial random effects, is modeled through a Conditional Autoregressive (CAR) framework, to account for spatial dependencies between zones. This makes it valuable tools for policymakers and researchers aiming to address spatial inequalities in various fields, including public health, urban development, and environmental management.

Spatial logistic regression provides several advantages in analyzing binary outcomes, particularly by incorporating spatial dependencies that traditional models often overlook. This approach enhances model fitness and predictive accuracy, facilitating the identification of high-risk areas through spatial clustering of central zone, northwestern zone, and southern, southeastern, Eastern and Mekelle zones of the Tigray region Ethiopia. It also uncovers spatial patterns and clusters, which can inform targeted public health interventions [18–20]. The analysis was conducted using R version 4.4.1, with weighing applied to adjust for non-proportional sampling. Both exploratory and confirmatory analyses were carried out, including descriptive statistics, visualisations, and inferential statistics. Predictors with p < 0.05 in the bivariate and multivariate analysis were incorporated into the final spatial logistic regression model.

## Results

### Socio-demographic characteristics of participants

The study participants are divided into two groups: host communities 2,069 (84.7%, 95% CI: [83.5%, 85.9%]) and internally displaced people 375 (15.3%, 95% CI: [14.1%, 16.5%]). The mean age of participants was 28 years, with a standard deviation of 0.05.

A significant portion of the participants have not completed formal education. Specifically, 835 individuals (34.2%, 95% CI: [32.5%, 36.0%]) are illiterate and only 201 individuals (8.2%, 95% CI: [7.3%, 9.1%]) hold a college degree or higher.

Geographically, most participants reside in rural areas, with 1,530 (62.6%, 95% CI: [60.9%, 64.3%]). The majority of 2,239 participants (91.6%, 95% confidence interval: [90.5%, 92.7%]), were married.

Access to resources was limited among the participants. For instance, 1,615 (66.1%, 95% CI: [64.4%, 67.8%]) lack electricity, and 1,373 (56.2%, 95% CI: [54.5%, 57.9%]) do not have access to public transportation. Media ownership is also low, with only 308 participants (12.6%, 95% CI: [11.3%, 13.9%]) owning a radio. In contrast, mobile phone ownership is more common, with 1,138 individuals (46.6%, 95% CI: [44.9%, 48.3%]) possessing a mobile phone. Television ownership is relatively low, as just 400 participants (16.4%, 95% CI: [15.1%, 17.7%]) have a television or detail see "Table 1."

### Sexual and reproductive health characteristics of participants

The study participants reported varying experiences regarding pregnancies and births. The majority had 2 pregnancies (21.7%), followed by one pregnancy (20.3%). The number of pregnancies decreases with higher counts: three pregnancies (17.2%), five (10.4%), 6 (6.5%), seven (4.7%), and 8 (5.2%).

In terms of births, most participants reported one birth (22.7%) or two births (22.6%), with subsequent numbers decreasing: three births (16.6%), four (11.9%), five (9.0%), six (5.7%), seven (3.8%), and eight (3.7%). Additionally, 4.1% have not given birth.

Regarding live births, the distribution is as follows: one live birth (23.2%), two (22.2%), three (16.6%), four (11.9%), five (8.5%), 6 (5.6%), seven (3.5%), and eight or more (3.2%). Notably, 5.3% of participants reported having no live births.

The majority, 2,328 (95.0%), reported zero stillbirths, while 95 (3.9%) had 1 stillbirth, 18 (0.7%) had 2, and 2 (0.08%) had 3, and 1 (0.04%) had 4 stillbirths.

Regarding abortions, 1,622 (66.4%) reported having 0 abortions in their lifetime. However, a notable portion has experienced abortions: 698 (28.6%) reported 1 abortion, 100 (4.1%) reported 2, 18 (0.7%) reported 3, 5 (0.2%) reported 4, and 1 (0.04%) reported 5 (Fig 1).

### Pregnancy intentions among participants

The study revealed varied pregnancy intentions among participants. A majority, 1,455 (59.5%), reported were actively trying to become pregnant at the time of conception. However, a significant portion, 738 (30.2%), expressed a desire to wait until later to have children. Additionally, 251 (10.3%) stated that they did not wish to have any more children when they become pregnant. The findings of this study indicate a diverse range of pregnancy intentions within the study population, highlighting that while many participants were intentionally seeking pregnancy, a notable number preferred to delay or avoid further pregnancies. For more details, see "Table 2".

Antenatal care (ANC) interruption: The prevalence of antenatal care (ANC) interruption was found to be 24.8% (95% CI: [23.1%, 26.5%]), indicating that nearly one in four women encountered difficulties in maintaining consistent ANC. The primary reasons for these interruptions included insecurity (45.3%, 95% CI: [43.1%, 47.5%]), health facility closures (39.8%, 95% CI: [37.6%, 42.0%]), and a lack of supplies and medicine for ANC (33.1%, 95% CI: [31.0%, 35.2%]).

**Table 1. Socio-demographics characteristics of participants Tigray region northern Ethiopia (N 2444).**

| Variables | Response | Frequency(n) | Percent (%) |
|---|---|---|---|
| Zone | South | 346 | 14.2 |
| | South-east | 233 | 9.5 |
| | Mekelle | 308 | 12.6 |
| | Eastern | 520 | 21.3 |
| | Central | 624 | 25.5 |
| | Northwest | 413 | 16.9 |
| Participant | Host | 2069 | 84.7 |
| | IDP* | 375 | 15.3 |
| Religion | Orthodox | 2369 | 96.9 |
| | Muslim | 75 | 3.1 |
| Residence | Rural | 1530 | 62.6 |
| | Semi-urban | 190 | 7.8 |
| | Urban | 724 | 29.6 |
| Educational status | Unable to read and write | 835 | 34.2 |
| | Able to read and write | 114 | 4.7 |
| | Elementary school | 754 | 30.9 |
| | Secondary school | 540 | 22.1 |
| | College and above | 201 | 8.2 |
| Marital status | Single | 41 | 1.7 |
| | Married | 2239 | 91.6 |
| | Divorced | 127 | 5.2 |
| | Widowed | 11 | 0.5 |
| | Separated | 26 | 1.1 |
| Access to electricity | Yes | 829 | 33.9 |
| | No | 1615 | 66.1 |
| Radio ownership | Yes | 308 | 12.6 |
| | No | 2136 | 87.4 |
| Television ownership | Yes | 400 | 16.4 |
| | No | 2044 | 83.6 |
| Mobile phone ownership | Yes | 1138 | 46.6 |
| | No | 1306 | 53.4 |
| Have been displaced | Yes | 1386 | 56.7 |
| | No | 1058 | 43.3 |
| Does the war impact your life | Yes | 1421 | 58.1 |
| | No | 1023 | 41.9 |
| Time taken round trip to health post | < 1 hour | 1583 | 64.8 |
| | >=1 hour | 861 | 35.2 |

*Internally displaced people

The spatial distribution of ANC interruptions across the Tigray region is as follows: Northwestern zone 22.8%, Central zone 29.1%, Eastern zone 17.7%, Southern zone 16.7%, Southeastern zone 6.4%, and Mekelle zone 7.3%. However, the Western zone of Tigray was not accessible for our study due to security concerns. For more details, see Fig 2.

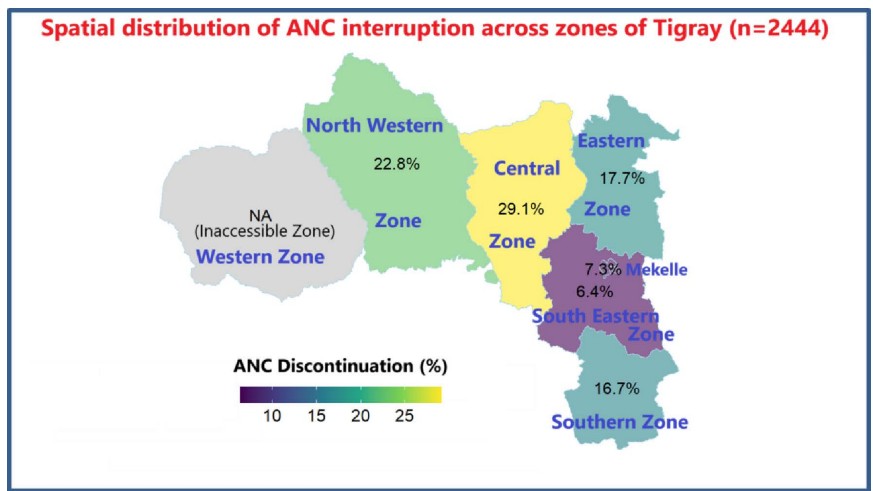

**Fig 1. Reasons for interruption of ANC care during the war from Pretoria November 2020 to August 2022 agreement of the Tigray region, Northern Ethiopia.**

**Table 2. War and ANC interruption of ANC care during the war to Pretoria November 2020 to August 2022 agreement of the Tigray region Northern Ethiopia (N-2444).**

| Predictors | Categories | ANC Interruption due to war | | $x^2$ | P-value | Predictors | Categories | ANC Interruption due to war | | $x^2$ | P-value |
|---|---|---|---|---|---|---|---|---|---|---|---|
| | | No(n = 1839) | Yes(n = 605) | | | | | No(n = 1839) | Yes(n = 605) | | |
| Sick Last 2 Weeks | Yes | 349 (14.3%) | 171 (7%) | 23.44 | <0.001 | HP Distance | <= 1 hour | 1214(49.7%) | 369(15.1%) | 5.03 | 0.025 |
| | No | 1490 (61.0%) | 434 (17.8%) | | | | >= 1 hour | 625(25.6%) | 236(9.7%) | | |
| Disabled War | Yes | 61(2.5%) | 25(1.0%) | 0.89 | 0.345 | Functional HP | Yes | 994(57.8%) | 309(18.0%) | 1.78 | 0.183 |
| | No | 177(72.7%) | 580(23.7%) | | | | No | 249(14.5%) | 93(5.4%) | | |
| Lost Family member in War | Yes | 49(2.0%) | 25(1.0%) | 3.34 | 0.068 | Functional HC | Yes | 56(37.3%) | 19(12.7%) | 38.51 | <0.001 |
| | No | 1790(73.2%) | 580(23.7%) | | | | No | 18 (12.0%) | 57(38.0%) | | |
| Rape victim HHs | Yes | 27(1.1%) | 7(0.3%) | 0.32 | 0.571 | Access to trans-port to HP/HC | Yes | 19 (3.2%) | 90 (15.3%) | 9.03 | 0.003 |
| | No | 1812(74.1%) | 598(24.5%) | | | | No | 153 (26.0%) | 326 (55.4%) | | |
| War Damage | Yes | 490(34.5%) | 174(12.2%) | 0.76 | 0.383 | Electricity Supply | Yes | 24 (23.5%) | 78 (76.5%) | 0.942 | 0.005 |
| | No | 543(38.2%) | 214(15.1%) | | | | No | 18 (24.0%) | 57 (76.0%) | | |
| War Looted | Yes | 94(6.6%) | 53(3.7%) | 11.38 | 11.38 | Radio in the HH | Yes | 24 (23.5%) | 78 (76.5%) | 0.942 | 0.005 |
| | No | 939(66.1%) | 335(23.6%) | | | | No | 18 (24.0%) | 57 (76.0%) | | |
| Displaced due to War | Yes | 1027(42.0%) | 359(14.7%) | 2.26 | 0.138 | TV in HH | Yes | 24 (23.5%) | 78 (76.5%) | 0.942 | 0.005 |
| | No | 812(33.2%) | 246(10.1%) | | | | No | 18 (24.0%) | 57 (76.0%) | | |
| War Impact in HH | Yes | 1033(42.3%) | 388(15.9%) | 11.85 | <0.001 | Mobile Ownership | Yes | 24 (23.5%) | 78 (76.5%) | 0.942 | 0.005 |
| | No | 806(33.0%) | 217(8.9%) | | | | No | 18 (24.0%) | 57 (76.0%) | | |

## Bivariate analysis: Impact of war on antenatal care (ANC) visits

The study found that 75.2% (1,839, 95% CI: [73.9%, 76.5%]) of participants did not postpone or interrupted their antenatal care (ANC) visits due to the ongoing war, while 24.8% (605, 95% CI: [23.5%, 26.1%]) reported interruptions. Among those with a sick family member in the last two weeks, 14.3% (349, 95% CI: [12.9%, 15.7%]) did not postpone their ANC visit, and 7.0% (171, 95% CI: [5.9%, 8.1%]) did Fig 3.

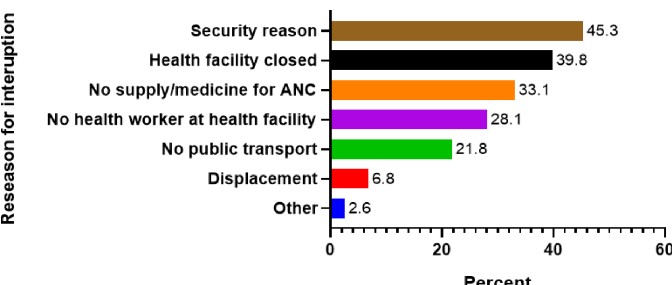

**Fig 2. Special distribution of ANC interruption across the zone of Tigray Region (n = 2444).**

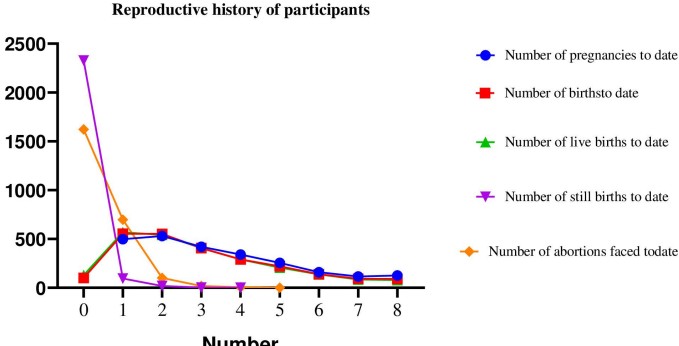

**Fig 3. The line graph shows the reproductive history of study participants of the interruption of ANC during the war in Tigray 2023.**

The war's impact was significant, with 96.5% (2,358, 95% CI: [95.4%, 97.6%]) reporting a family member was disabled; of these, 72.7% (1,778, 95% CI: [71.0%, 74.4%]) did not postpone their ANC visit, while 23.7% (580, 95% CI: [22.2%, 25.2%]) did. Additionally, 97.0% (2,370, 95% CI: [96.0%, 98.0%]) experienced the loss of a family member, with 73.2% (1,790, 95% CI: [71.5%, 74.9%]) continuing their ANC visits. A small fraction, 1.4% (34, 95% CI: [0.8%, 2.0%]), reported a woman in their household experiencing rape due to the war, with only 1.1% (27, 95% CI: [0.5%, 1.7%]) not postponing ANC visits.

Property damage was reported by 53.3% (757, 95% CI: [51.0%, 55.6%]) of participants, with 38.2% (543, 95% CI: [35.9%, 40.5%]) not postponing their ANC visits. Looting affected 89.7% (1,274, 95% CI: [88.5%, 90.9%]), with 66.1% (939, 95% CI: [64.5%, 67.7%]) continuing their visits. Destruction of property was noted by 27.3% (388, 95% CI: [25.6%, 29.0%]) of participants, where 20.3% (288, 95% CI: [18.7%, 21.9%]) did not postpone. Displacement impacted 56.7% (1,386, 95% CI: [54.9%, 58.5%]), with 42.0% (1,027, 95% CI: [40.3%, 43.7%]) not postponing their ANC visits.

In terms of distance to health posts, 64.8% (1,583, 95% CI: [63.0%, 66.6%]) reported a round trip of less than 1 hour, while 35.2% (861, 95% CI: [33.4%, 37.0%]) reported longer trips. Those who did not postpone their ANC visits were more likely to live closer to health facilities (49.7% vs. 15.1%).

Functionality of nearby health facilities was high, with 75.8% (1,303, 95% CI: [74.1%, 77.5%]) reporting nearby health posts as functional and 97.0% (2,371, 95% CI: [96.0%, 98.0%]) for health centers/hospitals. Access to public transport was reported by 43.8% (1,071, 95% CI: [41.9%, 45.7%]), with those who had not postponed ANC visits slightly more likely to have access (32.8% vs. 11.0%).

Regarding electricity supply, 33.9% (829, 95% CI: [32.1%, 35.7%]) reported having it, and those who did not postpone their ANC visits were somewhat more likely to have electricity (26.5% vs. 9.0%). Media ownership was limited, with 12.6%

(308, 95% CI: [11.3%, 13.9%]) owning a radio and 16.4% (400, 95% CI: [15.1%, 17.7%]) owning a television. Those who did not postpone their ANC visits were slightly more likely to own a radio (9.6% vs. 3.5%) or a TV (12.6% vs. 4.5%). Mobile phone ownership was similar between both groups, with 46.6% (95% CI: [44.9%, 48.3%]) owning a phone.

## Multivariate analysis: Parameter estimates from SLR

The Moran I test indicated significant spatial autocorrelation in ANC interruption (Moran's I = 0.546, p-value < 0.001), indicating the presence of spatial autocorrelation in the dependent variable, validating the need for spatial logistic regression. Incorporating spatial dependencies is crucial, as neglecting them in ordinary logistic regression may result in biased parameter estimates and inaccurate inferences. Model fitness improved substantially with the inclusion of the spatial term, as reflected by an increase in log-likelihood from −1200 to −1050 and a decrease in AIC from 2410 to 2110. The pseudo-R-squared increased from 0.55 to 0.72, demonstrating better explanatory power. Multicollinearity diagnostics showed all Variance Inflation Factors (VIFs) below 3, indicating no serious multicollinearity issues among predictors.

The spatial logistic regression model identified several key factors influencing the interruption of antenatal care (ANC) during the war. For every unit increase in distance from the health center, the log-odds of interrupting ANC increased by 0.001 (SE = 0.001, p < 0.001). Geographic differences also played a significant role; women living in the south-east zone had a 0.112 lower log-odds of interrupting ANC compared to those in the north-west zone (SE = 0.05, p = 0.02). Additionally, women in the eastern zone exhibited a 0.16 lower log-odds of interrupting ANC compared to the north-west zone (SE = 0.04, p = 0.001).

The number of children under five in the household was another essential factor, with each additional child increasing the log-odds of interrupting ANC by 0.08 (SE = 0.02, p < 0.001). Furthermore, households without a radio had a 0.08939 lower log-odds of interrupting ANC compared to those with a radio (SE = 0.03, p = 0.001). These results suggest that longer distances to health centers, residing in the north-west zone, having more young children, and not owning a radio were all associated with a greater likelihood of ANC interruption in the war-affected Tigray region of Ethiopia. In contrast, the analysis for Mekelle indicated an estimate of −0.04, a standard error of 0.07, and a z-value of −0.55, resulting in a p-value greater than 0.05, which shows no significant effect on ANC interruption compared to the north-west zone.

In contrast, the estimate for women in Mekelle was −0.038 (SE = 0.069, p > 0.05), indicating no statistically significant difference in ANC interruption compared to the north-west zone. These findings highlight the importance of geographic location, access to information, and household burden in shaping ANC service continuity in war-affected areas of Tigray, Ethiopia. Conversely, the eastern zone showed significantly lower odds of ANC interruption, with an estimate of −0.160 (SE = 0.050, p < 0.05), indicating better adherence to ANC in this zone as seen in Table 3.

## Discussion

The prevalence of antenatal care (ANC) interruption was found to be 605 (24.8%), indicating that nearly one in four women faced challenges in maintaining consistent ANC. The interruption was notably more severe among mothers residing in rural areas (370, 61.2%) compared to semi-urban (51, 8.4%) and urban (184, 30.4%) settings. A significant majority of those affected were host community (non displaced people)(519, 85.8%) rather than internally displaced mothers (86, 14.2%). Additionally, many participants were married (563, 91.1%).

Similar studies conducted in Syria in 2014–2017 have found that violent events significantly reduce the likelihood of pregnant women seeking and accessing essential healthcare services showed a negative correlation between bombardments and consultations, and ANC visits were reported; the RR was 0.95 (95% CI 0.94–0.97) and 0.95 (95% CI 0.93–0.98), respectively.

The fear and risks associated with ongoing conflicts deter pregnant women from seeking necessary care, increasing the vulnerability of both mothers and their unborn children to potential health complications [21]. Research in Yemen highlighted those pregnant women in remote, rural areas encountered substantial barriers to reaching ANC clinics due

Table 3. Spatial logistic regression model analysis shows the parameter estimates, standard errors, z-values, and p-values for the predictors of interruption of ANC during the war in Tigray 2023.

| Predictors | Estimate | Std. Error | Z-value | Pr(>|z|) |
|---|---|---|---|---|
| (Intercept) | 0.164124 | 0.152572 | 1.0757 | 0.28 |
| South-east zone | −0.11876 | 0.050501 | −2.3518 | 0.01** |
| Mekelle zone | −0.0377 | 0.068806 | −0.5479 | 0.58 |
| Eastern zone | −0.16452 | 0.048012 | −3.4266 | 0.001*** |
| Central zone | 0.038805 | 0.041975 | 0.9245 | 0.35 |
| Northwest zone | 0.035293 | 0.046268 | 0.7628 | 0.44 |
| Age | 0.000458 | 0.002425 | 0.189 | 0.85 |
| Number of under-five children | 0.083134 | 0.017295 | 4.8068 | 1.53 |
| Number of Pregnancies | 0.002781 | 0.008414 | 0.3305 | 0.74 |
| Family size | 0.002114 | 0.007472 | 0.2829 | 0.77 |
| Perceived wealth medium | 0.08881 | 0.128912 | 0.6889 | 0.49 |
| Perceived wealth poor | 0.060797 | 0.130133 | 0.4672 | 0.64 |
| Radio ownership | −0.08939 | 0.033425 | −2.6743 | 0.007** |
| Television ownership | −0.02766 | 0.051554 | −0.5365 | 0.59 |
| Mobile phone ownership | −0.04237 | 0.023887 | −1.7739 | 0.07* |
| Perceived food Security moderately secured | 0.03115936 | 0.047596 | 0.6547 | 0.51 |
| Perceived food Security insecure | 0.035233 | 0.051334 | 0.6863 | 0.49 |
| War impacted me | −0.04285 | 0.022561 | −1.8993 | 0.05* |
| Distance health post in minutes | −0.00044 | 0.000266 | −1.6706 | 0.09* |
| Distance of health center in minutes | 0.000353 | 0.000103 | 3.415 | 0.001*** |
| Access to transport health Center | −0.00869 | 0.029711 | −0.2925 | 0.76 |

*p < 0.05, **p < 0.01, ***p < 0.001

to infrastructure damage, lack of public transportation, and financial constraints [22]. Furthermore, studies in Afghanistan have shown that the Taliban's restrictions on women's movement and access to healthcare during conflicts severely limited pregnant women's ability to attend ANC appointments [23,24]. These challenges were particularly pronounced in rural areas compared to urban settings [25]. This is a critical concern since ANC is a well-established and cost-effective intervention that provides essential care to pregnant women from early pregnancy to delivery [26].

The high prevalence of interruptions in antenatal care (ANC) services during conflict situations contradicts the internal resolution of the World Health Organization (WHO, 2012), which stresses the importance of leadership in documenting attacks on health workers, facilities, and patients [27].

The results of the spatial logistic regression analysis in the war-affected Tigray region of Ethiopia revealed several key factors influencing ANC interruptions. For instance, the distance to health centers was a significant factor; for every unit increase in distance (in minutes), the log-odds of discontinuing ANC increased by 0.001 (SE = 0.0001, p < 0.001). This finding aligns with studies conducted in Syria, where ongoing conflict and insecurity significantly disrupted access to ANC services. This was compared by the time taken to reach health facilities both before the war and during the war, even if the distance is constant, the time taken in These two periods were different; this may be due to the security reasons, blockage, checkpoints here and there. Women reported facing barriers such as transportation difficulties, fear of checkpoints, and financial constraints when trying to reach health facilities [28,29]. Additionally, a study in Syria found that women living in besieged or hard-to-reach areas experienced much greater difficulty accessing ANC services due to transportation challenges, security concerns, and a lack of functioning health facilities [30].

Living in the north-west zone of the war-affected Tigray region was associated with a higher likelihood of interruption of antenatal care (ANC) this is similar with a study conducted during the Tigray war, 80 times of drone and aerial bombardment incidents were reported resulted in casualties in six zones and 24 districts in Tigray, Ethiopia. Nearly a third of the victims were children, and half of the casualties were females, including pregnant and lactating women. The Southern and Northwest zones caused the most casualties, Southeast zones and Mekelle. There were many civilian casualties from the airstrikes, which mostly occurred in civilian locations such as marketplaces, internally displaced persons (IDP) camps, residential areas, public transportation, villages, children's playgrounds, churches, mills, and hospitals. Women, children, and the elderly were disproportionately affected [31].This may be the blockage, movement from place to place, or severity of war may be higher in this area and because it is far from Mekelle the capital city of the Tigray region which is about 300kms, some of the humanitarian organisations were not reaching this area. This is also similar to studies conducted in Syria where bombardment and distant villages from the centers interrupted ANC service than those who are near the centers [21]. This is the fact that in conflict settings, healthcare access is severely compromised due to the destruction of facilities and the evacuation of medical personnel, leading to increased travel times and reduced service availability where health facilities are bombarded even the public transportation and other related facilities are hampered.

Conversely, while distance is a critical factor, other elements such as socio-economic status, education, and community support also plays a role in healthcare access and continuity in conflict-affected regions [32].

Also align with previous a study, which have consistently highlighted those geographic disparities, often exacerbated by conflict, plays a critical role in shaping healthcare utilization. For instance, in conflict-affected regions of Afghanistan, women in remote or insecure areas faced higher barriers to ANC attendance due to disrupted infrastructure and security concerns, a pattern that resonates with the higher likelihood of ANC interruption [33]. Additionally, households with a greater number of young children were more prone to discontinuation, as were those who did not own a radio. These findings underscore the unique challenges faced by pregnant women in war-affected regions, highlighting the need for improved access to health facilities, addressing transportation and security concerns, and implementing targeted interventions to support households with multiple young children.

Research from Yemen and the Democratic Republic of the Congo further illustrates the severe Impact of armed conflicts and civil wars on maternal healthcare access in sub-Saharan Africa. In Yemen, the escalation of the civil war led to the closure of many health facilities, significantly hindering pregnant women's ability to receive regular ANC [34,35]. Similarly, findings from the Democratic Republic of the Congo indicate that armed conflicts and displacement forced many pregnant women to miss critical ANC visits, resulting in poorer maternal and child health outcomes and also another study from Nepal indicates that women in conflict areas receive fewer ANC visits when there is armed conflict, which has a direct effect on maternal health outcomes.

[36–38]. Overall, the effects of civil wars on sub-Saharan Africa have severely compromised maternal and child health services [39].

The spatial logistic regression analysis highlighted distance to health centers as a significant factor; for every unit increase in distance (in minutes), the log-odds of discontinuing ANC increased by 0.001 (SE = 0.001, p < 0.001). Studies from Yemen reported substantial barriers for pregnant women in remote, rural areas trying to access ANC clinics, primarily due to infrastructure damage, lack of public transportation, and financial constraints [40,41]. Studies from Damage to the public health system caused by war-related looting or vandalism in the Tigray region of Northern Ethiopia, also showed that due to facility damage and medical personnel evacuation, access to healthcare is seriously jeopardized in crises, resulting in more extended travel times and fewer services available [42].Another study on the quality of health care service in a conflict setting showed that healthcare disruption in armed conflict creates significant obstacles to the provision of healthcare, such as medical supply shortages and access-hindering security threats [43].On the other hand, although distance is an important considerations, socioeconomic position, education, and community support are also important factors that affect healthcare continuity and access in areas affected by violence [44].

Geographic zones also played a critical role; women living in the south-east zone had a 0.11lower log-odds of discontinuing ANC compared to those in the north-west zone (SE = 0.05, p = 0.01). Similarly, women in the eastern zone exhibited a 0.16 lower log-odds of discontinuing ANC compared to the north-west zone (SE = 0.04, p = 0.001). Thus, living in the north-west zone was associated with a higher likelihood of ANC interruption in the Tigray region. This finding mirrors those from the Eastern Mediterranean region, where conflicts have disrupted health service utilization, especially in rural areas [45]. A study in the Democratic Republic of the Congo noted that pregnant women displaced to rural areas had significantly lower ANC centres, per cent of the population than those remaining in or fleeing to urban centers [46,47]. Furthermore, research in Afghanistan showed that the Taliban's restrictions on women's movement disproportionately affected rural women, leading to higher rates of missed ANC appointments compared to their urban counterparts [47–51].

These findings underscore the critical importance of addressing the unique challenges faced by pregnant women in war-affected regions, particularly those living in remote, rural areas with limited access to health facilities. Enhancing transportation, infrastructure, and security in these areas is essential for improving maternity healthcare utilization and outcomes.

Broader patterns in conflict-affected settings reveal that the uneven distribution of conflict intensity and damage to health infrastructure across different zones has led to varying impacts on pregnant women's ability to receive antenatal care (ANC). The findings from this study in Tigray align with another analysis indicating that, as of June 2021, only 27.5% of hospitals, 17.5% of health centers, 11% of ambulances, and none of the 712 health posts were functional [16].

The specific reasons for the higher interruption of antenatal care (ANC) in the north-west zone of Tigray remain unclear, highlighting the need for further contextual analysis to uncover the underlying factors contributing to this geographic disparity. Additionally, the number of children under five in a household significantly impacts ANC continuity; for each additional child, the log-odds of discontinuing ANC increase by 0.08 (SE = 0.02 p < 0.001). Parity is positively correlated with ANC interruptions, as demonstrated by demographic and health surveys in eight Sub-Saharan African (SSA) countries, particularly among women with five or more children. Our research indicates that mothers with more children are also more likely to cease receiving ANC care. This suggests that the demands of raising multiple children may hinder mothers' ability to prioritize their health care. The responsibilities associated with caregiving appear to overshadow the importance of regular ANC visits, potentially jeopardizing the health of both mothers and their children [52]. Another study conducted in Tanzania, also found that women with higher parity often relies on their previous pregnancy experiences, leading to a tendency to underutilize ANC services. This reliance on past experiences may create a false sense of security, causing these women to deem ANC less critical [53].However, the practical challenges of managing several young children further complicate their ability to seek necessary prenatal care, reinforcing the findings of the SSA survey and the present studies. Additionally, research from Liverpool studied the unique challenges faced by women experiencing multiple pregnancies. This study emphasized that increased anxiety and practical obligations hinder their capacity to prioritize ANC visits [54]. All those findings underscore as the number of children increased, which increases the interruption of care. This showed that mothers are overwhelmed by the care of children thus our ANC care contacts need to be tailored to support individualized care strategies to address the specific needs of women, ensuring that they receive adequate prenatal care.

This trend may reflect the additional caregiving responsibilities that pregnant mothers with multiple young children face, which can hinder their ability to prioritize their own ANC visits.

This may be severe in the condition of conflict where health facilities were closed due to war, health personnel displaced as shown by community-based study in armed conflict in Tigray showed that only 36.5%, respondents reported the use of optimal antenatal care (ANC) [55].

Moreover, household radio ownership emerged as an important factor; households without a radio had a 0.08939 lower log-odds of discontinuing ANC compared to those with a radio (SE = 0.033, p = 0.007). This suggests that access to media and information sources, such as radio, can positively influence ANC utilization, as mothers who

receive ANC-related information are more likely to act on it. study of antenatal care and skilled births in the fragile and conflict-affected situation of Burundi indicated that women who engage with media, including radio, are more likely to utilize healthcare services which is similar to the present study mothers who own radio did not interrupt ANC visits this may be the information disseminated through the radio may be indicating where the services is available and also information about security issues about war may be disseminated so us mothers know when to visit health facilities and when not to visit health facilities [56].

Another study conducted on Antenatal care utilization and determinants in fragile conflict-affected situations in northern Ethiopia showed that radio can foster community discussions around maternal health, increasing awareness and reducing stigma associated with seeking ANC in conflict zones thus in conflict affecting zones radio may be an appropriate massage or information transmission methods that can support health services utilization of communities in conflict affecting situations [55,57]. Another studies conducted in the political unrest in Nepal,Ethiopia and Burundi showed that in areas where physical access to healthcare is limited, radio can bridge the gap by informing women about available services and how to access them, potentially increasing ANC visits, this is another evidence that supports the present study, thus owning a radio in such conflict situations may get information that supports where and when to visit health facilities so that you get health services [36,57,58].

The Tigray study offers a more granular, zone-level analysis that complements zonal-level findings from other conflict-affected contexts, highlighting the importance of understanding subnational variations in healthcare access during conflicts. A recent study in the region indicates that the effect of airstrike and drone bombardment instances showed that nearly one-in-two of the causalities were female including pregnant and lactating women. |Out of the entire region most of the causalities occurred in southern and northwest zones followed by the Southeast zones and Mekelle [31]. Further research to unpack the underlying drivers of geographic disparities in ANC interruption in Tigray would be invaluable for informing more targeted interventions.

## Strength and limitation of the study

Spatial modeling offers significant strengths in studying antenatal care interruption, particularly through its ability to provide clear visual representations of geographic patterns. This clarity helps stakeholders identify areas with high rates of interruption, facilitating a better understanding of the spatial dynamics of healthcare access. Furthermore, these models allow researchers to incorporate a range of socio-economic, environmental, and conflict-related variables, leading to a comprehensive understanding of the factors influencing antenatal care. By pinpointing hotspots, health authorities can strategically allocate resources and design targeted interventions, enhancing the potential for improved access and outcomes in affected regions. Additionally, spatial modelling enables tracking changes over time, revealing trends related to ongoing conflicts or recovery efforts, which is crucial for understanding the dynamics of healthcare access in fluctuating environments. The insights gained from these models can also inform policymakers about the specific needs of affected populations, guiding the development of effective health service delivery strategies.

However, spatial modelling also faces notable limitations. In war-torn regions, the quality and availability of data can be compromised, leading to incomplete data like the western zone of the Tigray region not included because of security reasons or inaccessible at all. Overall, it results bias in the estimate.. Complex health outcomes are influenced by multiple interdependent factors that may not be fully captured, limiting the explanatory power of the models. Moreover, the assumptions and simplifications inherent in spatial modelling can oversimplify the intricate realities of healthcare access, potentially resulting in misinterpretations of the data. Rapid changes in conflict situations may outperform the data used in models, diminishing their relevance over time. Lastly, findings derived from specific regions may not be generalizable to other contexts, which restricts the broader applicability of results and may hinder the development of universal strategies for improving antenatal care access.

## Conclusion

The study identified the high prevalence of antenatal care (ANC) interruption, Key factors associated with the likelihood of interruption of ANC included greater distance from health centers, residing in the north-west zone compared to the southeast and eastern zones, having a higher number of children under five in the household, and household ownership of a radio.

## Recommendations

To address these findings, several strategies are recommended. First, improving access to health centers could be achieved by establishing more community-based ANC services or implementing transportation assistance programs. Second, targeted interventions are needed for high -risk north-west zone to specifically address the elevated rates of ANC interruption in this region. Third, providing additional resources and support, such as childcare assistance or conditional cash transfers could benefit households with multiple young children, helping to alleviate caregiving burdens.

Promoting the use of radio and other mass media channels is also crucial for disseminating information and encouraging ANC attendance, particularly in households with a radio. Additionally, further research is essential to unpack the underlying drivers of geographic disparities in ANC interruption in Tigray, which would inform more targeted and effective interventions. By addressing these factors, it may be possible to enhance the continuity of antenatal care and ultimately improve maternal and child health outcomes in the Tigray region.

The findings indicate significant interruptions in antenatal care (ANC) in Tigray, Driven by geographic disparities, household dynamics, and access to information. Specifically, the increased log-odds of interruptions with additional travel time highlight the urgent need for improved infrastructure and security. The lower log-odds of interruption in the southeast and eastern zones suggest that targeted interventions could be beneficial in areas most affected. Furthermore, the association between household dynamics and ANC interruptions emphasizes the importance of enhancing mass media health information dissemination to improve access to care. Therefore, further research on geographic disparities is essential to inform tailored strategies for improving ANC services in the region.

## Ethical considerations

Ethical clearance was approved by the Tigray Health Research Institute review board with reference number (THRI 4031/1099/15). A permission letter was obtained from the Tigray regional health bureau. Oral consent and volunteer participation was obtained from study participants after fully communicating the benefits and risks of participation in the study with a right to discontinue participation at any interview stage. Data confidentiality was maintained throughout the research process. In conducting our research with vulnerable populations in conflict zones, we prioritize several ethical considerations to ensure the integrity and safety of the study. Informed consent is fundamental; we ensure that participants fully understand the study's purpose, risks, and benefits by simplifying the language to Tigrigna and providing culturally relevant information. Safety and security are paramount, so we assess potential risks associated with participation and implement measures to mitigate them, safeguarding participants, researchers, and the community. We also maintain confidentiality by protecting participants' identities and personal information through anonymization techniques and secure data storage, particularly in sensitive contexts. Our approach is guided by cultural sensitivity, recognizing and respecting local norms and values, and actively engaging with communities to ensure our methods are appropriate. We strive for beneficence and non-maleficence, aiming to maximize benefits while minimizing harm, considering how our research outcomes can positively impact the community. Community engagement is crucial; we involve local stakeholders and organizations, such as health facility workers, in the data collection process to enhance the relevance and acceptance of our research. Additionally, we plan for post-study benefits, ensuring that findings are shared with participants and the community so they can directly benefit from the results. Finally, we have obtained approval from an ethics review board, acknowledging the unique challenges posed by conflict situations and reinforcing our commitment to ethical research practices.

## Acknowledgments

We extend our heartfelt gratitude to the Tigray Health Bureau (THB) for leading this integrated research and mobilizing technical staff for data collection. Lastly, we thank the participants, as well as the data collectors and supervisors, particularly those from the Mekelle University College of Health Sciences and Tigray Health Research Institute (THRI), for their commitment and integrity in generating quality data.

## Author contributions

**Conceptualization:** Assefa Ayalew Gebreslassie, Mussie Alemayehu.

**Data curation:** Assefa Ayalew Gebreslassie, Haftu Gebrehiwot, Ferehiwot Hailemariam, Tsegay Wellay, Adhena Ayalew, Znabu Hadush Kahsay.

**Formal analysis:** Assefa Ayalew Gebreslassie, Mussie Alemayehu, Haftu Gebrehiwot, Brhane Ayele, Tsegay Wellay, Znabu Hadush Kahsay, Liya Mamo, Afework Mulugeta Bezabh.

**Funding acquisition:** Melaku Abraha, Hayelom Kahsay, Tsegay Berihu, Ashenafi Asmelash, Afework Mulugeta Bezabh.

**Investigation:** Assefa Ayalew Gebreslassie, Mussie Alemayehu, Haftu Gebrehiwot, Brhane Ayele, Hailay Gebretnsae, Ferehiwot Hailemariam, Araya Abrha Medhanyie, Mohammedtahir Yahya, Melaku Abraha, Hayelom Kahsay, Tsegay Hadgu, Fana Gebreslassie, Asfawosen Aregay, Kiros Demoz, Mulugeta Tilahun, Mulugeta Woldu, Ataklti Gebrtsadik, Abraham Aregay Desta, Amanuel Haile, Rieye Esayas, Tsegay Berihu, Abrham Gebrelibanos, Tadele Tesfean, Ashenafi Asmelash, Afework Mulugeta Bezabh.

**Methodology:** Assefa Ayalew Gebreslassie, Mussie Alemayehu, Haftu Gebrehiwot, Brhane Ayele, Hailay Gebretnsae, Araya Abrha Medhanyie, Mebrahtu Kalayu Chekole, Tsegay Hadgu, Asfawosen Aregay, Kiros Demoz, Mulugeta Tilahun, Mulugeta Woldu, Ataklti Gebrtsadik, Abraham Aregay Desta, Amanuel Haile, Rieye Esayas, Tsegay Berihu, Abrham Gebrelibanos, Tadele Tesfean, Ashenafi Asmelash, Afework Mulugeta Bezabh.

**Project administration:** Mussie Alemayehu, Brhane Ayele, Araya Abrha Medhanyie, Mohammedtahir Yahya, Melaku Abraha, Hayelom Kahsay, Tsegay Hadgu, Amanuel Haile, Rieye Esayas, Tsegay Berihu, Ashenafi Asmelash, Afework Mulugeta Bezabh, Gebrehaweria Gebrekuristos.

**Resources:** Gebrehaweria Gebrekuristos.

**Supervision:** Assefa Ayalew Gebreslassie, Haftu Gebrehiwot, Brhane Ayele, Hailay Gebretnsae, Ferehiwot Hailemariam, Tsegay Wellay, Adhena Ayalew, Araya Abrha Medhanyie, Znabu Hadush Kahsay, Liya Mamo, Mebrahtu Kalayu Chekole, Reda Shamie, Mohammedtahir Yahya, Melaku Abraha, Tsegay Hadgu, Fana Gebreslassie, Asfawosen Aregay, Kiros Demoz, Mulugeta Tilahun, Mulugeta Woldu, Ataklti Gebrtsadik, Abraham Aregay Desta, Amanuel Haile, Rieye Esayas, Tsegay Berihu, Abrham Gebrelibanos, Tadele Tesfean, Afework Mulugeta Bezabh, Gebrehaweria Gebrekuristos.

**Validation:** Assefa Ayalew Gebreslassie, Haftu Gebrehiwot, Mebrahtu Kalayu Chekole, Abraham Aregay Desta, Tsegay Berihu.

**Visualization:** Haftu Gebrehiwot.

**Writing – original draft:** Assefa Ayalew Gebreslassie.

**Writing – review & editing:** Assefa Ayalew Gebreslassie, Mussie Alemayehu, Haftu Gebrehiwot, Brhane Ayele, Hailay Gebretnsae, Znabu Hadush Kahsay, Mebrahtu Kalayu Chekole, Reda Shamie, Fana Gebreslassie, Kiros Demoz, Afework Mulugeta Bezabh.

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
