## [Decision Letter · Decision Letter 0]

21 May 2025

PONE-D-25-08979Spatial patterns and predictors of antenatal care interruption in war-torn Tigray, northern Ethiopia: spatial modeling approachPLOS ONE

Dear Dr. Gebreslassie,

Thank you for submitting your manuscript to PLOS ONE. After careful consideration, we feel that it has merit but does not fully meet PLOS ONE’s publication criteria as it currently stands. Therefore, we invite you to submit a revised version of the manuscript that addresses the points raised during the review process.

We look forward to receiving your revised manuscript.

Kind regards,

Kahsu Gebrekidan, Ph.D.

Academic Editor

PLOS ONE

 [UNICEF, WHO, Amref Health Africa and UNFPA and Tigray regional heal bureau   and Mums for Mums (MfM) for all rounded support in ensuring a successful and productive integrated research data collection process we assure there is no any competing programming and any other support that influences the research output]. 

[We extend our heartfelt gratitude to the Tigray Health Bureau (THB) for leading this integrated research and mobilizing technical staff for data collection. Lastly, we thank the participants, as well as the data collectors and supervisors, particularly those from Mekelle University College of Health Sciences and Tigray Health Research Institute (THRI), for their commitment and integrity in generating quality data.]

[UNICEF, WHO, Amref Health Africa and UNFPA and Tigray regional heal bureau   and Mums for Mums (MfM) for all rounded support in ensuring a successful and productive integrated research data collection process we assure there is no any competing programming and any other support that influences the research output]. 

5. We note that your Data Availability Statement is currently as follows: [Data is available within the manuscript]

6. Please amend either the title on the online submission form (via Edit Submission) or the title in the manuscript so that they are identical.

7. Your ethics statement should only appear in the Methods section of your manuscript. If your ethics statement is written in any section besides the Methods, please move it to the Methods section and delete it from any other section. Please ensure that your ethics statement is included in your manuscript, as the ethics statement entered into the online submission form will not be published alongside your manuscript.

Reviewers' comments:

Reviewer's Responses to Questions

**Comments to the Author**

1. Is the manuscript technically sound, and do the data support the conclusions?

Reviewer #1: Yes

Reviewer #2: Yes

2. Has the statistical analysis been performed appropriately and rigorously? 

Reviewer #1: Yes

Reviewer #2: Yes

3. Have the authors made all data underlying the findings in their manuscript fully available?

Reviewer #1: Yes

Reviewer #2: Yes

4. Is the manuscript presented in an intelligible fashion and written in standard English?

Reviewer #1: Yes

Reviewer #2: No

5. Review Comments to the Author

Reviewer #1: Title: Spatial patterns and predictors of antenatal care interruption in war-torn Tigray, northern Ethiopia: a spatial modeling approach

I am happy to review this manuscript, which is well-written for the “Spatial patterns and predictors of antenatal care interruption in war-torn Tigray, northern Ethiopia: spatial modeling approach” and has brought some important findings. However, I have encountered some typos error in many instances. That is proofread is needed.

• Why you used mixed methods?

• In Abstract: Result Part: A result should be interpreted well and written to the nearest value. It seems vague.

• Conclusion of abstract, should be written again, especially recommendation part. Theoretically, recommendation should be emanated from result. Which result leads to this recommendation “Improving infrastructure and security, implementing targeted interventions, enhancing mass media health information dissemination, and further research on geographic disparities are essential.”?

• General comment: In introduction, it seems good. Please, rephrase and conceptualize in your own words. Basically, your introduction also should focus on a) what we already know from the literature, b) what new concepts this study can address that make a distinct contribution beyond the prior studies? People want to see the effect of war in the ANC use? Previous distribution should be written well.

• Methods part: Good but describe study area in connection to your title in detail.

• Perhaps, I am expecting from you special distribution of the problem in the map. Geographical map is needed to show the existence of the problem at the end of the result as a result or at the end of the introduction as the concept of indication.

• Measurement and variables should be written in detail.

• Ethical considerations part: Ethical approval and consent to participate part should follow the Helsinki declaration of human participants while you are writing ethics.

• In results part, good to narrate in this way. However, the study should show new insights and should enhance practical relevance while clarifying the scientific concept.

In Discussion part:

• Please do not repeat result (Numbers). Rather the implication of each association should be clearly explained.

• Under the discussion section, you have to start a discussion by summarizing the main findings.

• Possible scientific justification is required for each association whether it agrees or not with other studies.

Good luck!

Reviewer #2: The research project's title is scientifically sound, and I believe it will significantly contribute to the body of knowledge in maternal health. However, the document is not written in standard English and requires revision. Additionally, some areas need substantial changes and improvements. All comments and suggestions are tracked changes.

Comment # 1 Introduction: Explain these new Guidelines. It does not make sense to readers.

Comment #2 Materials and methods: Please clarify whether this research utilised secondary data from the "Evaluating" survey, "Cost-Effectiveness in Post-Conflict Settings in Tigray, Ethiopia," or relied on primary data.

Comment#3 Result: Please specify the model's fitness and describe the methods employed to assess multicollinearity.

Comment #4 Result: Indicate the variables with a P-value of less than 0.05 or significant.

Comment #5 Discussion: What is the data from the study conducted in Syria?

Comment # 6 Discussion: How can you compare the distance to health facilities with the impact of conflict and war on the interruption of antenatal care services?

Comment # 7 Discussion: Justify why this particular zone has higher odds of ANC interruption. Compare and contrast with similar findings.

Comment #8 Discussion: Correlate these findings with similar studies.

Comment # 9 Discussion: You should move this sentence to the recommendation section.

Comment # 10 Discussion: You need to establish a connection between the increased distance to healthcare facilities in war-torn areas and the discontinuation of the ANC program in rural regions. Rural areas often suffer from inadequate infrastructure, particularly in terms of roads, which hinders access to health facilities even in peaceful places. Consider how you can uniquely frame this argument to support your position.

Comment # 11 Discussion: The argument lacks clarity. Does the North-West zone have more rural kebeles compared to the South-West zone? How can you formulate an argument regarding the rural population in this context?

Comment # 12 Discussion: Move this paragraph to the recommendations section.

Comment # 13 Discussion: You need to provide reference findings.

Comment # 14 Discussion: You need to provide reference findings.

Comment # 15: Add the strengths and limitations of the study.

Comment # 16: Declare any conflict of interest.

6. PLOS authors have the option to publish the peer review history of their article (what does this mean? ). If published, this will include your full peer review and any attached files.

**Do you want your identity to be public for this peer review?** For information about this choice, including consent withdrawal, please see our Privacy Policy .

Reviewer #1: **Yes: ** Feleke Doyore Agide

Reviewer #2: No

---

## [Author Response · Author response to Decision Letter 1]

12 Jun 2025

Response to editor and Reviewers

PLOS ONE

Thank you for the opportunity to revise our manuscript titled "Spatial Patterns and Predictors of Antenatal Care Interruption in War-Torn Tigray, Northern Ethiopia: Spatial Modeling Approach Manuscript ID: PONE-D-25-08979. We appreciate the constructive feedback from you and the reviewers. Below, we address each comment in detail.

Response: our funding had not grant number because the budget was utilizing through perdiem .

Thank you for stating the following financial disclosure:

[UNICEF, WHO, Amref Health Africa and UNFPA and Tigray regional heal bureau and Mums for Mums (MfM) for all rounded support in ensuring a successful and productive integrated research data collection process we assure there is no any competing programming and any other support that influences the research output]. The funders had no role in study design, data collection and analysis, decision to publish, or preparation of the manuscript

Please state what role the funders took in the study. If the funders had no role, please state:

Response thank you for your comment we have revised as you suggested

UNICEF, WHO, Amref Health Africa and UNFPA and Tigray regional heal bureau and Mums for Mums (MfM) for all rounded support in ensuring a successful and productive integrated research data collection process we assure there is no any competing programming and any other support that influences the research output. The funders had no role in study design, data collection and analysis, decision to publish, or preparation of the manuscript

In the research on antenatal care interruption in Tigray, we maintain a strong commitment to avoiding conflicts of interest. We ensure that our funding sources do not create biases, as we do not have any conflicts with organizations that might have vested interests in the outcomes, such as government agencies UNICEF, WHO, Amref Health Africa and UNFPA and Tigray regional heal bureau and Mums for Mums (MfM with specific agendas. Our collaborative relationships with local health authorities and NGOs are structured to prevent any conflicts, as these entities do not have a stake in the research findings that could influence our work. Additionally, the researchers do not have personal ties to the region or its healthcare systems that could compromise the interpretation of data or reporting results. We are also vigilant against publication bias, ensuring that we do not face conflicts that would pressure us to publish only positive outcomes aligned with funding interests, which could distort the integrity of our research. Ethical considerations are paramount; we adhere to rigorous ethical guidelines to address the unique challenges of working with vulnerable populations in conflict zones, recognizing the dilemmas that may arise. To mitigate potential conflicts, we prioritize transparency by disclosing all funding sources and potential conflicts in our publications. We are committed to an independent review process, welcoming input from independent experts in study design and data analysis to ensure objectivity. Finally, we uphold ethical oversight by obtaining approval from relevant review boards Tigray health research institute (THRI) as mentioned above to safeguard participant welfare, reinforcing our dedication to conducting responsible and ethical research

[We extend our heartfelt gratitude to the Tigray Health Bureau (THB) for leading this integrated research and mobilizing technical staff for data collection. Lastly, we thank the participants, as well as the data collectors and supervisors, particularly those from Mekelle University College of Health Sciences and Tigray Health Research Institute (THRI), for their commitment and integrity in generating quality data.]

[UNICEF, WHO, Amref Health Africa and UNFPA and Tigray regional heal bureau and Mums for Mums (MfM) for all rounded support in ensuring a successful and productive integrated research data collection process we assure there is no any competing programming and any other support that influences the research output].

Response: thanks we have removed from the acknowledgements section

5. We note that your Data Availability Statement is currently as follows: [Data is available within the manuscript]

Response: Thank you we will submit the data set if our manuscript will be accepted

6. Please amend either the title on the online submission form (via Edit Submission) or the title in the manuscript so that they are identical.

Response: Thank you for your comment we have made identical

7. Your ethics statement should only appear in the Methods section of your manuscript. If your ethics statement is written in any section besides the Methods, please move it to the Methods section and delete it from any other section. Please ensure that your ethics statement is included in your manuscript, as the ethics statement entered into the online submission form will not be published alongside your manuscript.

Response ;it is only in the methods sections

Response: I have added six references

31-Mesfin, B., Demise, A. M., Shiferaw, M., Gebreegziabher, F., & Girmaw, F. (2023). The Effect of Armed Conflict on Treatment Interruption, Its Outcome and Associated Factors Among Chronic Disease Patients in North East, Amhara, Ethiopia, 2022. Patient Related Outcome Measures, 14, 243–251. https://doi.org/10.2147/prom.s388426

32-Tessema, Z. T., & Akalu, T. Y. (2020). Spatial Pattern and Associated Factors of ANC Visits in Ethiopia: Spatial and Multilevel Modeling of Ethiopian Demographic Health Survey Data. Advances in Preventive Medicine, 2020, 4676591. https://doi.org/10.1155/2020/4676591.

33-Akseer, N., et al. (2020). Progress in maternal and child health: How has Afghanistan fared in the last decade? The Lancet Global Health, 8(3), e351–e360.

42-Asah, M., & Moses, A. L. (2025). The Effects of Armed Conflict on Healthcare in Nigeria – A Scoping Review. International Journal of Research and Scientific Innovation, XI(XV), 883–900. https://doi.org/10.51244/ijrsi.2024.11150067p

43-Ikpongifono, U. (2024). Enhancing Healthcare Access in Conflict Zones: Identifying Challenges and Proposing Solutions. Public Health Open Access, 8(1). https://doi.org/10.23880/phoa-16000286

44-Mavole, J. (2023). Protection and Maintenance of Healthcare Services for Civilians’ Well-Being in Conflict Affected Areas: Comparative Analysis of The African Situation. European Journal of Health Sciences, 9(2), 1–17. https://doi.org/10.47672/ejhs.1507

General Comments

5. Review Comments to the Author

Reviewer #1: Title: Spatial patterns and predictors of antenatal care interruption in war-torn Tigray, northern Ethiopia: a spatial modeling approach

I am happy to review this manuscript, which is well-written for the “Spatial patterns and predictors of antenatal care interruption in war-torn Tigray, northern Ethiopia: spatial modeling approach” and has brought some important findings. However, I have encountered some typos error in many instances. That is proofread is needed.

• Why you used mixed methods?

Response: in the integrated survey it was used to collect the qualitative one but I did not use the qualitative because it is mixed up with others like COVID19 socio-behavioral drivers, other maternal health services that is why I did not use for ANC interruption.

• In Abstract: Result Part: A result should be interpreted well and written to the nearest value. It seems vague.

Response :Great I made like this

This finding underscores significant accessibility challenges posed by security checkpoints and conflict-related dangers. Geographic zone location significantly affected ANC interruptions. Women in the southeast and eastern zones experiencing lower log-odds of interruption compared to the northwest zone. Indicating zonal disparities in access to care.

Moreover, the number of children under five years old in a household was associated with a 0.083134 increase in log-odds of ANC interruption (SE = 0.017295, p < 0.001). This suggests that larger family sizes may contribute to greater challenges in accessing ANC services.

Additionally households without a radio had a 0.08939 lower log-odds of interruption (SE = 0.033425, p = 0.007489), indicating the importance of access to information to mitigate ANC interruption.

• Conclusion of abstract, should be written again, especially recommendation part. Theoretically, recommendation should be emanated from result. Which result leads to this recommendation “Improving infrastructure and security, implementing targeted interventions, enhancing mass media health information dissemination, and further research on geographic disparities are essential.”?

Response: Thank you this is really good here I amend like this

Conclusion

The findings indicate significant interruptions in antenatal care (ANC) in Tigray, Driven by geographic disparities, household dynamics, and access to information. Specifically, the increased log-odds of interruptions with additional travel time highlight the urgent need for improved infrastructure and security. The lower log-odds of interruption in the southeast and eastern zones suggest that targeted interventions could be beneficial in areas most affected. Furthermore, the association between household dynamics and ANC interruptions emphasizes the importance of enhancing mass media health information dissemination to improve access to care. Therefore, further research on geographic disparities is essential to inform tailored strategies for improving ANC services in the region.

• General comment: In introduction, it seems good. Please, rephrase and conceptualize in your own words. Basically, your introduction also should focus on a) what we already know from the literature, b) what new concepts this study can address that make a distinct contribution beyond the prior studies? People want to see the effect of war in the ANC use? Previous distribution should be written well.

Response: I tried to incorporate it is acceptable

• Methods part: Good but describe study area in connection to your title in detail.

Response: Study area: The study was conducted in Tigray region, Northern Ethiopia. Tigray borders on the north by Eretria, on the west by Sudan, on the south by Amhara, and on the east by Afar. The region has a total population of 7,969,000 []. The region’s health system consisted of two specialized comprehensive hospitals, 14 general hospitals, 24 primary hospitals, 231 health centers, and 743 health posts. The survey was conducted in selected 19 districts of the six zones of the region (excluding western zone for security reasons). Data were collected from July 1 to 30, 2023

• Perhaps, I am expecting from you special distribution of the problem in the map. Geographical map is needed to show the existence of the problem at the end of the result as a result or at the end of the introduction as the concept of indication.

Response:

At the end of introduction

At the end of results were added and at the introduction

The spatial distribution of ANC interruptions across the Tigray region is as follows: Northwestern zone 22.8%, Central zone 29.1%, Eastern zone 17.7%, Southern zone 16.7%, Southeastern zone 6.4%, and Mekelle zone 7.3%. However, the Western zone of Tigray was not accessible for our study due to security concerns. For more details, see Fig 1.

• Measurement and variables should be written in detail.

Response

Study Design: Community based cross sectional study

Measurements and variables

Dependent variable; antenatal care interruption:

Measured as a binary variable indicating whether ANC services were interrupted (1) or not interrupted (0) during the conflict period. This was assessed through participant’s interviews in selected households of mothers who were following ANC visit during the conflict and interrupted their care of different reasons.

Independent Variables

Independent variables include a range of socio-economic, geographical, and conflict-related factors:

Socio-economic factors: age of mother, education level, occupation, marital status,

Geographical Factors: zone of residence, distance to health facility: Measured in kilometers and time taken to double trip from the home to the nearest health facility providing ANC services. Accessibility index a composite measure reflecting road conditions, transportation availability, and geographic barriers, security. Conflict-related Factors; displacement status whether the mother or household has been displaced due to conflict (yes/no).Perceived security a qualitative measure based on survey responses regarding feelings of safety when accessing health services. Loss, damage of property and nearby persons who care for them.

Data collectio

---

## [Decision Letter · Decision Letter 1]

26 Jun 2025

PONE-D-25-08979R1Spatial patterns and predictors of antenatal care interruption in war-torn Tigray, northern Ethiopia: spatial modelling approachPLOS ONE

Dear Dr. Gebreslassie,

Thank you for submitting your manuscript to PLOS ONE. After careful consideration, we feel that it has merit but does not fully meet PLOS ONE’s publication criteria as it currently stands. Therefore, we invite you to submit a revised version of the manuscript that addresses the points raised during the review process.

We look forward to receiving your revised manuscript.

Kind regards,

Kahsu Gebrekidan, Ph.D.

Academic Editor

PLOS ONE

Journal Requirements:

Reviewers' comments:

Reviewer's Responses to Questions

**Comments to the Author**

1. If the authors have adequately addressed your comments raised in a previous round of review and you feel that this manuscript is now acceptable for publication, you may indicate that here to bypass the “Comments to the Author” section, enter your conflict of interest statement in the “Confidential to Editor” section, and submit your "Accept" recommendation.

Reviewer #1: All comments have been addressed

Reviewer #2: (No Response)

2. Is the manuscript technically sound, and do the data support the conclusions?

Reviewer #1: Yes

Reviewer #2: Yes

3. Has the statistical analysis been performed appropriately and rigorously? 

Reviewer #1: Yes

Reviewer #2: Yes

4. Have the authors made all data underlying the findings in their manuscript fully available?

Reviewer #1: Yes

Reviewer #2: (No Response)

5. Is the manuscript presented in an intelligible fashion and written in standard English?

Reviewer #1: Yes

Reviewer #2: No

6. Review Comments to the Author

Reviewer #1: (No Response)

Reviewer #2: You have addressed the majority of the comments and suggestion. However, the manuscript did not write in format of academic English. So, I recommend proofreading. I also have one comment. Comment #1: Please separate the conclusion and recommendation into distinct paragraphs. First, summarize the significant findings, followed by the recommendations. Currently, they are intermixed.

7. PLOS authors have the option to publish the peer review history of their article (what does this mean? ). If published, this will include your full peer review and any attached files.

**Do you want your identity to be public for this peer review?** For information about this choice, including consent withdrawal, please see our Privacy Policy .

Reviewer #1: **Yes: ** Feleke Doyore Agide

Reviewer #2: No

---

## [Author Response · Author response to Decision Letter 2]

4 Jul 2025

Response to Editors and reviewers

Kahsu Gebrekidan, Ph.D.

Academic Editor

PLOS ONE

Dear Dr. Kahsu Gebrekidan and esteemed reviewer,

I hope this message finds you well. I am writing to submit a revised version of our manuscript titled Spatial Patterns and Predictors of Antenatal Care Interruption in War-Torn Tigray, Northern Ethiopia: Spatial Modeling Approach (PONE-D-25-08979R1: Spatial patterns and predictors of antenatal care interruption in war-torn Tigray, northern Ethiopia: spatial modelling approach) in response to the valuable comments and questions provided by the editor and reviewers.

We sincerely appreciate the reviewers' thorough evaluations and insightful feedback, which have greatly enhanced the quality of our work. Below, I summarize our responses to each of the reviewers' comments, along with an outline of the revisions made in the manuscript. We have attached the revised manuscript, a version with track changes, and our one-on-one responses in the system.

Regards

Assefa Ayalew

Mekelle,Tigray ,Ethiopia .

Thank you so much for reviewing our work

1. One of the comments from our reviewers were to separate conclusion and recommendation, thank you so much accepted and we separated as the reviewer recommendation

2. The other comment was proof reading just we read and read by other co-authors too.

Response to editor

The issues raised by the editor was about referencing thank you so much the following references were replaced by other proper references just the replaced once are in the manuscript .

1. WHO, UNICEF, UNFPA and TheWorld Bank. Trends inMaternalMortality: 2000 to 2017WHO. Geneva:WHO (2019).

2. O’hare BAM, David PS. First do no harm: the impact of recent armed conflict on maternal and child health in Sub-Saharan Africa. J R Soc Med. (2007) 100:564–70. doi: 10.1258/jrsm.100.12.564

3. Akseer N, Wright J, Tasic H, Everett K, Scudder E, Amsalu R, et al. Women, children and adolescents in conflict countries: an assessment of inequalities in intervention coverage and survival BMJ Global Health. (2020) 5:e002214. doi: 10.1136/bmjgh-2019-002214

4. Abay ST, Gebre-Egziabher AG. Status and associated factors of birth registration in selected districts of Tigray region, Ethiopia. BMC international health and human rights. 2020 Dec;20:1-0.

5. Stamidis K, Bologna L, Losey L. CORE Group Polio Project (CGPP) Final Evaluation Report 2017.

6. Asah, M., & Moses, A. L. (2025). The Effects of Armed Conflict on Healthcare in Nigeria – A Scoping Review. International Journal of Research and Scientific Innovation, XI(XV), 883–900. https://doi.org/10.51244/ijrsi.2024.11150067p

7. Ahmadi A. Assessing Participation in Women's Development Projects in Afghanistan 2017

8. Owen, D. J., Wood, L., & Neilson, J. P. (2004). Antenatal care for women with multiple pregnancies: the Liverpool approach. Clinical Obstetrics and Gynecology, 47(1), 263–271. https://doi.org/10.1097/00003081-200403000-00026

---

## [Decision Letter · Decision Letter 2]

8 Jul 2025

Spatial patterns and predictors of antenatal care interruption in war-torn Tigray, northern Ethiopia: spatial modelling approach

PONE-D-25-08979R2

Dear Mr. Assefa,

We’re pleased to inform you that your manuscript has been judged scientifically suitable for publication and will be formally accepted for publication once it meets all outstanding technical requirements.

Kind regards,

Kahsu Gebrekidan, Ph.D.

Academic Editor

PLOS ONE

Additional Editor Comments (optional):

Reviewers' comments:

Reviewer's Responses to Questions

**Comments to the Author**

1. If the authors have adequately addressed your comments raised in a previous round of review and you feel that this manuscript is now acceptable for publication, you may indicate that here to bypass the “Comments to the Author” section, enter your conflict of interest statement in the “Confidential to Editor” section, and submit your "Accept" recommendation.

Reviewer #2: All comments have been addressed

2. Is the manuscript technically sound, and do the data support the conclusions?

Reviewer #2: Yes

3. Has the statistical analysis been performed appropriately and rigorously? 

Reviewer #2: Yes

4. Have the authors made all data underlying the findings in their manuscript fully available?

Reviewer #2: Yes

5. Is the manuscript presented in an intelligible fashion and written in standard English?

Reviewer #2: Yes

6. Review Comments to the Author

Reviewer #2: The manuscript was edited very well. It is scientifically sound and methodological rigorous research. I support the acceptance.

7. PLOS authors have the option to publish the peer review history of their article (what does this mean? ). If published, this will include your full peer review and any attached files.

**Do you want your identity to be public for this peer review?** For information about this choice, including consent withdrawal, please see our Privacy Policy .

Reviewer #2: No

---

## [Editor Report · Acceptance letter]

PONE-D-25-08979R2

PLOS ONE

Dear Dr. Gebreslassie,

I'm pleased to inform you that your manuscript has been deemed suitable for publication in PLOS ONE. Congratulations! Your manuscript is now being handed over to our production team.

Kind regards,

on behalf of

Dr. Kahsu Gebrekidan

Academic Editor

PLOS ONE